## [Decision Letter · Decision Letter 0]

12 May 2025

Response to Reviewers
Revised Manuscript with Track Changes
Manuscript
**Journal Requirements:**

1. Please send a completed 'Competing Interests' statement, including any COIs declared by your co-authors. If you have no competing interests to declare, please state "The authors have declared that no competing interests exist". Otherwise please declare all competing interests beginning with the statement "I have read the journal's policy and the authors of this manuscript have the following competing interests:"

**Additional Editor Comments (if provided):**

Following rigorous editorial evaluation, we acknowledge receipt of peer review feedback from two domain-specific experts who were meticulously selected for their subject matter authority. While limited in quantity, the quality and applicability of these reviews meet established scholarly standards for initiating the revision phase. We therefore recommend formally transmitting the complete review documentation to the author team to commence manuscript refinement. Additionally, should we receive further reviewer feedback, we will promptly notify your team.

Best wishes,

**Reviewers' Comments:**

**Comments to the Author**

1. Does this manuscript meet PLOS Digital Health’s publication criteria?

Reviewer #1: Yes

Reviewer #2: Partly

2. Has the statistical analysis been performed appropriately and rigorously?

Reviewer #1: Yes

Reviewer #2: Yes

3. Have the authors made all data underlying the findings in their manuscript fully available (please refer to the Data Availability Statement at the start of the manuscript PDF file)?

Reviewer #1: Yes

Reviewer #2: Yes

4. Is the manuscript presented in an intelligible fashion and written in standard English?

Reviewer #1: Yes

Reviewer #2: Yes

Reviewer #1: This manuscript works on a highly important and timely issue related to the propagation of anti-LGBTQIA+ bias through the clinical use of common commercial LLM's. The models chosen are a bit antiquated, but this weakness is balanced by a scarcity of literature elsewhere on the subject otherwise. It works well as a good foundation for additional work to build out more sophisticated bias evaluation frameworks for reasoning models in the future. In the current study the authors used 38 prompts developed in transparent collaboration with experts in medicine and LGBTQIA+ representation.

The use of dual axes was innovative and added another dimension for a more nuanced discussion of the results. The quantitative results were fairly descriptive rather than rigorous, but the qualitative results had adequate level of detail. and the mixed-methods approach built out a more complete picture. Minor areas of clarification should also include elaboration on how were the prompts derived, selected, and validated. The word "inappropriate" is a bit subjective and one might also note the careful limitation that identifying these qualitative examples can sometimes reinforce the biases sought to be removed unless navigated carefully. A brief word more in the discussion on over-anchoring and sycophantic behaviors would also elevate the conclusions, but this is not critical.

The authors also included practical next steps rather than shying away to mere problem identification, including high effort approaches such as prompt engineering or more objective retrieval methods. I recommend accepting for publication with minor revisions. The editor may consider whether some of the more detailed information in the supplemental files would be better served instead to include in the primary text.

Reviewer #2: I congratulate the authors for their courage and attention to one aspect of a broader and very important area of medical LLM research—quantifying, evaluating, and potentially mitigating bias to promote equity in LLM-based medicine.

Introduction

* Regarding the phrase “obtain information on potential medical treatments,” I would suggest elaborating further by including differential diagnoses, indications, and general health information, not only treatments.

* In the second part of this paragraph, I would expand the literature review about bias in LLMs. Specifically, the authors should cite more evidence about bias against gender and historically marginalized groups. I recommend citing an important recent large-scale study published in Nature Medicine (DOI: 10.1038/s41591-025-03626-6), as well as other recent relevant works.

* The statement at the start of the second paragraph is not correct. As mentioned earlier, the cited study specifically investigates biases against LGBTQIA+ communities on a large scale. A relevant preprint in PubMed (DOI: 10.1101/2025.03.04.25323396) also addresses this. Although I agree the topic remains under-researched, the current wording should be corrected.

* In general, I think the introduction should better establish and present the current literature.

Methods

* I suggest including more simplified examples when presenting the prompting strategy. The example provided—“e.g., LLM response tells LGBTQIA+ patient to ‘be honest’ about their symptoms”—is useful. Adding similar short, clear examples for each type of prompt would greatly enhance readability. Although detailed examples are in the supplementary materials, this alone is insufficient. Additionally, listing key information and instructions used for generating the case scenarios would help readers understand the methods clearly.

* The authors should also include details about the “infrastructure” in the manuscript itself. For example: how the models were run, when they were run, who executed the prompts, how the answers were extracted, and how the process was validated.

* I appreciate the grouping and prompting strategy, but these should be explained in greater detail in the manuscript. Many readers might not refer to the supplementary materials.

Results

* The sentence “This refusal did not occur disproportionately for prompts with LGBTQIA+ patients, but seemed triggered by specific words linked to LGBTQIA+ identity and health (e.g., vaginoplasty, puberty blockers)” belongs more in the discussion section than in results. The authors should first present an overview of results, quantitative and qualitative, without commentary. Reporting this directly with relevant numbers would be clearer.

* My first impression from the results is that the inaccuracies were primarily based on sexual orientation or identity. However, several inaccuracies described (e.g., recommendations on PAP screening age) are not identity-based. This difference between general inaccuracies and identity-based biases should be clearly distinguished and elaborated. For instance, in the phalloplasty example, it would be useful if the prompt explicitly stated there were no procedural complications. Without this specificity, the LLM’s inaccurate generalizations about eGFR might not directly indicate bias. I recommend reconsidering this.

* Concerning the puberty-blocker example: although the response may be incomplete or neutrally phrased, deeming it “inaccurate” might be excessive. Was incompleteness consistently treated as inaccuracy? I would appreciate the authors’ thoughts on this.

* Overall, I have mixed feelings. The prompting methods, specificity of issues addressed, and evaluation seem appropriate, and the use of inclusive language indicates expertise in LGBTQIA+ healthcare. Yet the presentation is somewhat unclear. As a reviewer, even after multiple readings, including supplementary materials, I found it difficult to form a clear view. Readers might similarly struggle. This work is important, particularly given previous larger-scale studies (such as the mentioned Nature Medicine publication) establishing biases towards LGBTQIA+ communities (especially in mental health recommendations). Thus, the value of this paper lies in the nuance it provides. I suggest rethinking the results’ presentation, clearly distinguishing between gender- or sexual-identity-related biases and other inaccuracies. Comparing cases labeled with identity characteristics versus unlabeled cases would offer more robust evidence of bias. Making the results section more direct and purely reporting (avoiding discussion) would enhance clarity and impact.

* In the paragraph beginning “Models frequently inappropriately created and justified differential,” I suggest emphasizing the differences in recommendations and wording choices between labeled and unlabeled prompts. Such comparisons provide clearer evidence of biased outcomes, likely reflecting biases from pre-training data.

Discussion

* The sentence starting with “In the only study to” is outdated, as discussed earlier. Since then, several studies have appeared, including a systematic review in the Journal of Equity and the previously mentioned large-scale Nature Medicine study, alongside recent preprints and smaller studies.

* Limitations should be explicitly detailed. For instance, the study’s scale is small; it does not evaluate newer models (e.g., Gemini Flash 2.0, Claude 3.5/3.7) or open-source models. Additionally, some bias instances flagged might be controversial or subjective due to the evaluator’s interpretation. Furthermore, since each prompt was run only once, accuracy and consistency were not assessed through repeated or rephrased runs.

* I recommend making the conclusions more nuanced and cautious, clearly aligned with the study's specific scope, scale, and noted limitations.

In general, I enjoyed reading the paper and find it valuable. However, it requires thorough revision, particularly in presenting results more clearly, specifically, and accurately, without mixing in discussion. The recent literature should be more comprehensively discussed, methodological details better explained in the manuscript itself, and conclusions adjusted accordingly. These are merely suggestions from a fellow researcher who acknowledges the significance of this important research area.

Best.

**Do you want your identity to be public for this peer review?** For information about this choice, including consent withdrawal, please see our Privacy Policy

Reviewer #1: **Yes: ** Catherine Bielick

Reviewer #2: No

**Figure resubmission:****Reproducibility:**To enhance the reproducibility of your results, we recommend that authors of applicable studies deposit laboratory protocols in protocols.io, where a protocol can be assigned its own identifier (DOI) such that it can be cited independently in the future. Additionally, PLOS ONE offers an option to publish peer-reviewed clinical study protocols. Read more information on sharing protocols at https://plos.org/protocols?utm_medium=editorial-email&utm_source=authorletters&utm_campaign=protocols

---

## [Decision Letter · Decision Letter 1]

18 Aug 2025

Evaluating anti-LGBTQIA+ medical bias in large language models

PDIG-D-24-00500R1

Dear Dr. Daneshjou,

We're pleased to inform you that your manuscript has been judged scientifically suitable for publication and will be formally accepted for publication once it meets all outstanding technical requirements.

Within one week, you'll receive an e-mail detailing the required amendments. When these have been addressed, you'll receive a formal acceptance letter and your manuscript will be scheduled for publication.

An invoice for payment will follow shortly after the formal acceptance. To ensure an efficient process, please log into Editorial Manager at https://www.editorialmanager.com/pdig/ click the 'Update My Information' link at the top of the page, and double check that your user information is up-to-date. For billing related questions, please contact billing support at https://plos.my.site.com/s/.

Kind regards,

Xiaoli Liu, PhD

Academic Editor

PLOS Digital Health

Additional Editor Comments (optional):

Dear Dr. Roxana Rose Daneshjou,

Both reviewers recommended acceptance. Congratulations! Great work! We look forward to this work reaching its relevant readership soon.

Best wishes,

Dr. Liu

Reviewers' comments:

Reviewer's Responses to Questions

**Comments to the Author**

Reviewer #2: All comments have been addressed

publication criteria?

Reviewer #2: Yes

3. Has the statistical analysis been performed appropriately and rigorously?

Reviewer #2: Yes

4. Have the authors made all data underlying the findings in their manuscript fully available (please refer to the Data Availability Statement at the start of the manuscript PDF file)?

Reviewer #2: Yes

5. Is the manuscript presented in an intelligible fashion and written in standard English?

PLOS Digital Health does not copyedit accepted manuscripts, so the language in submitted articles must be clear, correct, and unambiguous. Any typographical or grammatical errors should be corrected at revision, so please note any specific errors here.

Reviewer #2: Yes

Reviewer #2: I really congratulate the authors on their efforts to revise the manuscript.

I have enjoyed reading the paper as I noted the first time, but it was somewhat complicated and not easy to read, engage with, and fully understand from the first time. Now I think it is way better, more organized, and constructed.

No further comments from my end.

Keep up the good work!

**Do you want your identity to be public for this peer review?** For information about this choice, including consent withdrawal, please see our Privacy Policy

Reviewer #2: Yes: Mahmud Omar
